# Collagen Family as Promising Biomarkers and Therapeutic Targets in Cancer

**DOI:** 10.3390/ijms232012415

**Published:** 2022-10-17

**Authors:** Laura Necula, Lilia Matei, Denisa Dragu, Ioana Pitica, Ana Neagu, Coralia Bleotu, Carmen C. Diaconu, Mihaela Chivu-Economescu

**Affiliations:** 1Department of Cellular and Molecular Pathology, Stefan S. Nicolau Institute of Virology, 030304 Bucharest, Romania; 2Faculty of Medicine, Titu Maiorescu University, 040441 Bucharest, Romania

**Keywords:** collagens, cancer, targeted therapy, extracellular matrix, biomarkers

## Abstract

Despite advances in cancer detection and therapy, it has been estimated that the incidence of cancers will increase, while the mortality rate will continue to remain high, a fact explained by the large number of patients diagnosed in advanced stages when therapy is often useless. Therefore, it is necessary to invest knowledge and resources in the development of new non-invasive biomarkers for the early detection of cancer and new therapeutic targets for better health management. In this review, we provided an overview on the collagen family as promising biomarkers and on how they may be exploited as therapeutic targets in cancer. The collagen family tridimensional structure, organization, and functions are very complex, being in a tight relationship with the extracellular matrix, tumor, and immune microenvironment. Moreover, accumulating evidence underlines the role of collagens in promoting tumor growth and creating a permissive tumor microenvironment for metastatic dissemination. Knowledge of the molecular basis of these interactions may help in cancer diagnosis and prognosis, in overcoming chemoresistance, and in providing new targets for cancer therapies.

## 1. Introduction

Cancer represents one of the most important causes of death worldwide. According to GLOBOCAN data, almost 19.3 million new cancer cases were reported in 2020, and the five most frequent cancers, excluding non-melanoma skin cancer, are breast, lung, colorectum, prostate, and stomach cancer. Despite the advances in cancer therapy, this high incidence rate has been estimated to increase [1]. The worldwide cancer burden is expected to reach 28.4 million cases by 2040, which would be an increase of 47% from 2020. Cancer mortality statistics reported almost 10.0 million cancer deaths, and lung cancer remains the leading cause of cancer death, followed by colorectal, liver, stomach, and female breast cancers, in 2020 [2]. This high rate of mortality is explained by the large number of patients diagnosed at advanced stages when the treatment is often useless [1].

Several tumor markers specific for a certain type of cancer or different types of cancers have proven their usefulness in cancer screening, diagnosis, and prognosis as well as response to therapy or early recurrent/metastatic disease detection. Tumor markers, such as carcinoembryonic antigen (CEA), prostate-specific antigen (PSA), alpha-fetoprotein (AFP), carbohydrate antigen 19-9 (CA-19-9), carbohydrate antigen 125 (CA-125), beta subunit of human chorionic gonadotropin (b-hCG), lactate dehydrogenase (LDH), and chromogranin A (CgA), are used in abdominal and pelvic cancers. However, these tumor markers show limitations in terms of sensitivity (<40%) and specificity and cannot be used alone since there are several benign diseases or other conditions that can cause a false-positive increase in these markers, while some of the cancer patients do not show modifications [3]. For example, even if the CA 125 serological test is used in the case of ovarian cancer suspicion, this test has limited utility in the early diagnosis of ovarian cancer due to its low sensitivity and low specificity in premenopausal women [4]. Moreover, a study on 768 gastric cancer patients showed that 15.4% of patients had increased pre-operative CEA levels, and only 8.7% had increased CA19-9 levels [5], suggesting that these markers are not useful for early detection.

In an attempt to overcome this issue, several studies have shown that an association of CEA, CA19-9, and CA72.4 can be used as triple markers with increased sensitivity of up to 62% in gastric cancer [6].

AFP is the most commonly used biomarker for hepatocellular carcinoma, but the level of AFP is also increased in other benign liver diseases. Therefore it is recommended to use a combination of AFP testing with other factors, such as platelets and age, CEA and CA-19-9, and microRNAs, for hepatocellular carcinoma screening [7].

PSA is currently used as a biomarker for prostate cancer screening and diagnosis, and a low baseline PSA indicates a low risk of developing prostate cancer [8]. The German Institute for Quality and Efficiency in Health Care reported this year that PSA testing increases prostate cancer diagnoses and reduces long-term disease-specific mortality, but it also results in overdiagnosis and treatment-related harm [9].

Recently, a study on a large cancer patient cohort has highlighted the use of CA19.9 and CEA as prognostic biomarkers, their level being associated with shorter recurrence-free survival (RFS) and overall survival (OS) [10]. In conclusion, the results suggest the utility of these tumor markers in the detection of recurrence and postoperative surveillance rather than early detection of new cancers [11].

Cancer treatment is usually represented by a combination of chemo/radiation therapy, surgery, in some cases, hormonal therapy, and targeted therapy including immunotherapy, but each of these approaches has its side effects [1]. Moreover, the heterogeneity of tumor cells remains the main obstacle in obtaining an effective cancer treatment focused only on cancer cells [12]. It is very clear that we need to invest in the development of a cancer screening that can detect pre-neoplastic lesions, or at least early-stage cancer, allowing the treatment to improve the patient outcome and prolong survival. Colorectal and cervical cancer are just two examples that demonstrate the efficacy of cancer screening and that the removal of pre-neoplastic lesions, such as colon polyps and cervical intra-epithelial neoplasia, can significantly reduce mortality [13,14,15].

However, high mortality rates and the lack of biomarkers with high sensitivity and specificity support the need for the development of new non-invasive biomarkers for the early detection of cancer and new therapeutic targets for better health management.

## 2. Collagen Family, Structure, and Function

Collagens (COL) are a large group of proteins that share a common structural feature: the presence of at least one triple-helical domain made of three polypeptide chains twisted around one another. Collagen family members are abundantly found in the extracellular matrix (ECM) and are a component of the cell membrane. Besides critical structural roles, organization, and shape of tissues, they are involved in cell attachment, proliferation, and migration processes [16].

The three polypeptide chains that associate with the primary structure of collagens are α chains, and they might form homotrimers when the chains are identical (such as in COL2) or heterotrimers when different chains assemble (COL9) [16]. The α chain domains are rich in (Gly-X-Y) amino acid sequences where X and Y are proline and hydroxyproline [17].

The presence of alternative promoters, mRNA splicing, different isoforms for each collagen type, as well as post-translational modifications are responsible for the molecular diversity of collagen family members [18].

To date, 28 collagen types with different structures, localizations, and biological functions have been identified. For example, COL1, the most common type of collagen, is a fibrillar protein found in the skin, tendons, ligaments, blood vessels, bone, lungs, and heart. It is involved in physiological processes such as bone mineralization, hemostasis, and angiogenesis but also in cancer and atherosclerosis [19]. Mutations in COL1 genes are responsible for connective tissue and bone diseases such as Ehlers–Danlos syndrome and osteogenesis imperfecta [20]. COL4, an important component of the basement membrane, forms a complex network involved in interactions with cells thus having significant roles in adhesion, migration, survival, proliferation, and differentiation processes [21].

Based on their supramolecular assembly, several groups of collagen have been described: fibrillar collagens, fibril-associated collagens with interrupted triple helices (FACITs), network-forming collagens, transmembrane collagens, multiplexins (endostatin-producing collagens), anchoring fibrils, and beaded-filament-forming collagen [22].

Although fibroblasts are specialized cells whose main function is the synthesis of collagen, other types of cells, such as smooth muscle cells and epithelial cells, can synthesize collagen too.

Collagen synthesis is a multi-step process that implies co- and post-translational enzymatic modifications of great importance for its structure and function [23]. The synthesis begins inside the fibroblast nucleus where the α-chain genes are transcribed into mRNA. The polypeptide chain is synthesized as a prepro-α-chain from the mRNA. Then, in the rough endoplasmic reticulum lumen, the signal peptide is enzymatically cleaved on the N-terminal, and the pro-α-chain is released. The pro-α-chain undergoes post-translational enzymatic modification that includes hydroxylation of proline and lysine residues and glycosylation of the selected hydroxyl groups on hydroxylysine [24,25]. Three chains assemble to form a procollagen monomer and are joined together by the disulfide bonds formed at the propeptide regions (the N and C terminal ends of the peptide chains) [26]. The precursor is transported to the Golgi apparatus where it undergo further modifications (oligosaccharides are added), and secretory vesicles are assembled in order to be exocytosed. Outside the cell, the collagen peptidases cleave the pro-peptide regions, converting procollagen to tropocollagen. The monomers spontaneously arrange to form the fibrils. Lysyl oxidase creates covalent bonds between collagen monomers that stabilize the fibrillar structure [27]. Collagenases, enzymes belonging to the metalloprotease group, and cathepsins are responsible for degrading collagen molecules [22].

## 3. Collagen Family and Cancer

### 3.1. Relation with ECM/EMT, Regulator of the Tumor-Associated Immune Cells, Tumor Infiltration, Tumor Angiogenesis

In the last two decades, it has become very clear that a tumor is not only composed of malignant cells but also of the extracellular matrix, each of them playing an important role in disease progression and metastasis [28]. The epithelial to mesenchymal transition (EMT) and the extracellular matrix (ECM) have a collaborative effect in the metastatic process, implying increased collagen stabilization and deposition in the ECM of metastatic tumor tissues as a direct consequence of amplified collagen gene expression [29].

The ECM, the non-cellular component of tissues and organs [30], is mainly composed of water and biomolecules such as polysaccharides and proteins. The proteins constitute the complement of the ECM, or the mammal “core matrisome,” which comprises almost 300 proteins: 43 collagen subunits, 36 or so proteoglycans, and around 200 glycoproteins [30,31]. COL4 is the main component of the ECM [28,30]. COL4A1, in a pathological condition, was identified as one of the “hub genes” implicated in the regulation of different types of cancers such as breast cancer, bladder cancer, and gastric cancer [32,33,34]. Depending on the context, COL4 can have both pro- and anti-tumor effects, but in most studies, it was correlated with poor survival, most likely due to a higher risk of developing distant metastases [35]. In vitro studies on endothelial cells suggested that tumstatin, derived from the α3 chain of type IV collagen, and canstatin, derived from the α3 chain of type IV collagen, can inhibit angiogenesis by inhibiting human endothelial cell migration and can present proapoptotic activity [36,37,38]. However, in several types of cancer such as breast, pancreas, gastric, and colorectal cancers, fragments of collagen four have been identified as possible prognostic markers correlated with increased invasiveness and poor survival, most likely due to a higher risk of developing distant metastases [39,40,41].

The components and organization of the ECM differ depending on different tissues; these tissue-specific properties are implicated in determining tissue function [42]. Particularly, the organization, distribution, and abundance of the “core matrisome” are important in malignant transformation and also in tumor progression and metastasis [43]. For example, COL12, secreted by cancer-associated fibroblasts (CAFs), alters COL1 organization and creates a pro-invasive microenvironment that supports metastatic dissemination (Figure 1) [44]. CAFs originate from fibroblasts and mesenchymal stem cells, and they are the central players, and the functional regulators, present in the tumor mesenchyme which usually includes fibroblasts, immune cells, blood vessels, and the extracellular matrix [45,46].

Tumor desmoplasia is a specific cancer condition involving chronic inflammation. As reviewed in [43], there are some initial biological steps involved in malignant transformation. One of the first events is the remodeling of the basement membrane (thinner, laminin-111 lower expression) in order to invade the parenchyma. Additionally, collagen deposition increases, a process of progressive linearization, and the thickening of interstitial collagen were observed, with the formation of a tumor-specific ECM that is collagen-rich and with increased stiffness [47]. The diameter of collagen fibers is decreased by proteoglycan content, and because of the dehydration that occurs, the linearization of the COL1 fibers significantly enhances the tumor metastatic potential, favoring the migration of the tumor cells into the circulation [48]. Tumor epithelial cells produce collagenases, enzymes that are involved in the degradation of COL1 and that degrade the stroma. Furthermore, matrix metalloproteinases cause proteolysis of the ECM components with MMP-1 involved in the degradation of COL1 fibers, MMP-2 involved in the induction of cell migration, and MMP-3 involved in the cell apoptosis process [49]. The chronic inflammation specific to the process of malignant transformation is also supported by peripheral monocytes that migrate into the tumor microenvironment and that, in the presence of certain stimuli, differentiate into tumor-associated macrophages (TAMs). Stimuli are usually represented by chemokines, cytokines, and growth factors. Additionally, similar to CAFs, TAMs secrete proteases that alter collagen organization [50]. Besides the secretion of proteases, the contribution of TAMs in the reorganization of the ECM also consists of the production of chemokines, cytokines, growth factors, metalloproteinases, and pro-angiogenic factors that are directly involved in the modulation of the immune response in the processes of proliferation, invasion, and metastasis [51]. In addition, some studies demonstrate that a high-density ECM affects T-cell proliferation and its ability to kill cancer cells [52].

The EMT represents a fundamental process in embryonic development that is normally dormant in the adult organism, but which was also associated with tumor invasion and metastasis. In the epithelial to mesenchymal transition, epithelial cells lose the expression of epithelial markers. The mechanisms by which the EMT stimulates cell migration and cellular transformation were studied extensively in certain types of cancer, with ECM molecules implicated, and growth factors. COL1 has been shown to promote EMT. One of the pathways by which the EMT is aberrantly activated is through COL1 mediated phosphorylation of IκB in an ILK-dependent manner with the eventually increased expression of the Snail, LEF-1 transcription factors, and E-cadherin-reduced expression [53].

### 3.2. Plasma Levels of Collagens in Cancer Patients

The need to identify reliable circulating biomarkers that will allow early detection and/or to predict and monitor the response to treatment in cancer patients remains one of the priorities in the field of oncology. The detection of biomarkers in the blood of patients with solid tumors by using less invasive methods may have a significant clinical impact. In this context, the molecules derived from the tumor-microenvironment are of particular interest, especially collagens which represent the most abundant proteins of the ECM, and they were shown to be involved in tumorigenesis [54,55], cancer progression, and metastasis [56].

The plasma levels of different types of collagens were found to be elevated in various malignancies, but for the moment, only a few molecules that belong to this family can be considered as possible biomarkers for early cancer detection. For example, a recent study suggested that the evaluation of COL3 and MMP-1 levels can be used for the early detection of hepatocellular carcinoma [57]. Similarly, in the case of breast cancer, the evaluation of the circulating level of COL11A1, COL10A1, and COMP (collagen oligomeric matrix protein) can discriminate between a malign and a benign disease [58].

However, there are many studies that sustain the association between carcinogenesis and the modification in the circulating expression of the collagen family. High levels of circulating COL4 were reported in patients with primary breast cancer [59], while a more recent study suggested that COL4 could be used as a potential biomarker of metastatic breast cancer [60]. In colorectal cancer, it has been shown that elevated circulating COL4 is associated with liver metastases [41] and might also be used as a prognostic biomarker in patients with colorectal cancer [61]. A similar increase in plasma levels of COL10 has been reported in several malignant tumors. Elevated COL10A1 levels were associated with an advanced tumor stage and with poor survival [62] in gastric cancer patients, whereas in colon cancer, COL10A1 could be used to detect both adenoma lesions and invasive cancer [63].

More than that, several studies have shown that serum levels of collagen degradation products are elevated in cancer patients in comparison to healthy individuals. In a recent study, Willumsen et al. evaluated the biomarker potential of four fragments of COL6 in sixty-five serum samples from cancer patients. The data showed that C6Ma1 (MMP-generated neo-epitope on the a1 chain) and C6Ma3 (MMP-generated neo-epitope on the a3 chain) were significantly elevated in most cancer types compared to healthy controls and indicate that measuring the fragments of COL6 in serum could serve as a potential cancer biomarker [64].

Another study reported that an increase in the levels of circulating COL1 and COL3 degradation products (COL1 C-terminal telopeptide (ICTP) and COL3 N-terminal telopeptide (IIINTP)) was associated with poor survival in patients with head and neck squamous cell carcinoma (HNSCC) [65]. Additionally, collagen remodeling biomarkers (matrix metalloprotease (MMP)-degraded COL1 (C1M), COL3 (C3M), COL4 (C4M), and a pro-peptide of COL3 (PRO-C3)) were evaluated in patients with pancreatic ductal adenocarcinoma (PDAC) in a phase 3 clinical trial. The results showed that elevated serum levels of collagen degradation fragments in pre-treatment patients, reflecting an altered state of the ECM/collagen formation, were associated with poor survival outcomes in PDAC patients [66].

### 3.3. Collagen Tissue Levels in Cancer Patients

Significant advances in cancer biology were acquired in the last few years using bioinformatics analysis which highlighted deregulated genes and pathways involved in cancer development. Among the most frequently mentioned genes, the collagen family and other molecules of the tumor extracellular matrix always stood out and were associated with a poor prognosis in solid cancers [67,68,69].

In the remodeling steps during tumor growth, the normal ECM collagens are substituted with a tumor-specific collagen profile [70]. Increased expression of different collagens was associated with tumor progression and a poor prognosis in various types of epithelial cancers including colorectal [71], esophagus [72], glioma [73], gastric [74,75], head and neck [76], lung [77], ovarian [78], pancreatic [79], renal [80], breast [81,82], and cervical cancer [83]. The profile of different collagen types expressed in various solid cancers is presented as a heatmap in Figure 2. As indicated, the most frequently up-regulated collagens in solid cancers are those belonging to the COL1, COL3, COL4, COL6, and COL18 types.

Collagens regulate cancer cells’ polarity, migration, and signaling in the tumor microenvironment [84], but the protein levels of each collagen and its mechanism of action differ among cancer types. Recently, more and more studies are associating collagen overexpression with angiogenesis, invasion, and drug resistance in poor overall survival [85,86].

## 4. The Role of Collagens in Cancer: From Genetics to Targeted Therapy

Collagen organization and abundance, as part of the tumor extracellular matrix (ECM), are relevant features of solid tumors and are deeply involved in their progression. Several types of collagens were identified as being overexpressed in various tumors, and consequently, their role in tumor progression was enquired through gene manipulation models. At the same time, these investigations are expected to elucidate possible molecular mechanisms that may be manipulated through targeted therapy.

According to some in vitro functional studies, collagens are essential in supporting cancer cell proliferation as well as their migration and invasion capacities (Table 1). Different collagens, such as COL4A5, for example, may be responsible for efficient tumor angiogenesis, with their knockdown significantly reducing endothelial cell proliferation and migration as well as decreasing ERK phosphorylation but also impairing the tubule formation capability of endothelial cells [88].

More importantly is the fact that some collagens are differentially expressed in a normal and a tumoral context, and this feature can be exploited in selective therapy.

Some studies demonstrated that COL11A1 expression is absent in benign pathological conditions such as premalignant lesions, hyperplasia, fibrosis, cirrhosis, pancreatitis, and inflammatory bowel disease [89,90]. However, COL11A1 is well known to be upregulated in several cancers with epithelial cells, with a median fold change of 2.54, increasing up to 5.3 in breast cancer, followed by mesothelioma and pancreatic adenocarcinoma [91,92]. As such, COL11A1 was proposed as a tumor marker for the prognosis of breast, glioma, head and neck, lung, colorectal, esophagus, gastric, ovarian, pancreatic, salivary gland, and renal cancers [86,93].

COL11A1 promotes cell proliferation and inhibits cell apoptosis by activating Akt in cancer cells and was demonstrated to be involved in chemoresistance. The COL11A1/Akt/CREB axis exerts anti-apoptotic effects on cancer cells and protects tumors from Gemcitabine (GEM)-induced apoptotic cell death by modulating the function of the BAX/BCL-2 signaling node COL11A1/Akt, disturbing the BCL-2/BAX balance, inhibiting cytochrome c (Cyt-C) release, and binding Apaf-1/procaspase-9/Cyt-C, which suppresses the apoptotic program and induces GEM resistance in pancreatic cancer cells [94].

COL11A1 is known to be involved in many signaling pathways. According to several in vitro studies, Akt, TGF-beta1, B-myb, and Gli1 can up-regulate the transcription of COL11A1 in pancreatic [94], lung [95], and ovarian cancer [96]. The high expression of COL11A1 also induces the expression of molecules such as Twist1 and MMP3, which are related to drug resistance and invasion of cancer [97].

Different studies showed that collagens are regulated by the TGF-β signaling pathway that regulates the ECM, mediates the EMT [98,99], and are implicated in PI3K/AKT pathway activation [32,94]. Papanicolaou et al. evaluated the functional role of COL12, secreted by cancer-associated fibroblasts (CAFs), in primary breast tumor and metastasis formation and determined that COL12 is implicated in the spatial organization of COL1 fibrils in tumors, generating pro-invasive microenvironments [44].

It is important to note that collagen structural organization is still not completely understood. The different structures of the collagen fibers have been discriminating categories associated with the suppression of tumor growth or on the contrary, with proliferation and invasion [100].

**Table 1 ijms-23-12415-t001:** In vitro studies focused on collagen family.

Collagen Type	Cancer Type	Effect	Reference
**Collagen loss-of-function effect in vitro**			
COL1A1	gastric adenocarcinoma	inhibits cell proliferation	[101]
COL1A2	esophageal carcinoma	inhibits cell proliferation as well as reduces migration and invasion, changes the protein content of p-Akt and vimentin (S1)	[102]
COL3A1	glioma	significantly reduces proliferation (on day 4) and colony formation as well as reduces glioma cell migration	[103]
COL4A1	gastric adenocarcinoma	inhibits the proliferation	[101]
	pancreatic adenocarcinoma	reduces cell growth and migration, increases apoptosis	[104]
	lung cancer	reduces phosphorylation of Akt and Src	[88]
COL4A2	triple-negative breast cancer	inhibits proliferation by arresting the cell cycle in the G2 phase and inducing apoptosis, inhibits migration	[105]
COL4A5	lung cancer	reduces cell proliferation and anchorage-independent cell growth, reduces the phosphorylation levels of ERK and Akt	[88]
COL5A1	glioblastoma	decreases cell proliferation, inhibits migration and invasion	[106]
	breast cancer	decreases cell viability as well as migration and invasive capacities	[98]
	lung adenocarcinoma metastasis	increases apoptosis and inhibits proliferation and cell invasion capability	[107]
COL10A1	colorectal cancer	inhibits cell proliferation, reduces migration and invasion ability, suppresses EMT	[108]
	breast cancer	enhances proliferation and clone-forming ability, migration, and invasion capability	[109]
COL11A1	ovarian cancer	inhibits cell proliferation, invasion capacity, and anchorage-independent cell growth; reduces MMP3 gene expression and activity	[99]
	pancreatic cancer	decreases cell proliferation and gemcitabine resistance, increases apoptosis by reducing BCL-2 and increasing BAX and cleaved-caspase-3/9 expressions, inhibits p-BAX (Ser184) and decreases Akt (Ser473) and CREB (Ser133) phosphorylation.	[94]
COL13	breast cancer	reduces tumor sphere formation, reduces the number of invasive branches, and inhibits invasive growth in 3D culture; reduces the velocity and the distance of cell migration	[110]
COL23A1	lung carcinoma	Inhibits anchorage-independent growth, affects cell morphology, and decreases adhesive capability; decreases protein expression of OB-cadherin, b-catenin, a-catenin, g-catenin, vimentin, and galectin-3	[111]
**Collagen gain-of-function effect in vitro**			
COL1A1	gastric adenocarcinoma	promotes proliferation	[101]
COL4A1	gastric adenocarcinoma	promotes proliferation	[101]
COL10A1	colorectal cancer	increases migration and invasion ability, promotes EMT	[108]
	breast cancer	enhances proliferation and clone-forming ability, migration, and invasion capability	[109]
COL13	breast cancer	increases 3D colony size and invasiveness; significantly increases cell migration velocity and distance; enhances mammosphere formation; and increases TGF-β signaling, at least partially, through the β1 integrin pathway	[110]
COL17	breast cancer	decreases proliferation, clonogenicity, and growth; reduces the spheroid size and proliferation; reduces Ki67 expression; and deactivates the AKT/mTOR signaling pathway by inhibiting the phosphorylation of AKT, mTOR, p70S6K, and 4EBP1	[112]
COL23A1	non-small cell lung cancer	increases adhesive capability	[111]

In addition to the in vitro studies, there are also some in vivo studies conducted on animal models that underline the role of collagens in promoting tumor growth and creating a permissive tumor microenvironment for metastatic dissemination (Table 2). This is due to a secretion of thinner collagen fibrils and/or by altering fibril organization, modifying tumor stiffness, and inducing a pro-invasive microenvironment in tumors [44] or by the modulation of the immune microenvironment [70].

Through in vivo gene modulation experiments, it was also observed that collagens are involved in the development of drug resistance by inhibiting the penetration of the drugs into the cancer tissue. Thus, several collagens, such as COL1, COL3, COL5, COL12, and COL17, were associated with cytostatic drug resistance in ovarian cancer [118,119]. Interestingly, the ablation of collagens in human cancer models overcomes the resistance to therapy, improving the delivery and efficacy of anticancer treatments. For example, the knockdown of COL18A1 improved the efficacy of EGFR/ERbB-targeting therapies, abolishing drug resistance [117].

## 5. Conclusions

In this review, we provided an overview of the collagen family as promising biomarkers and of how they may be exploited as therapeutic targets in cancer. Collagen family tridimensional structure, organization, and functions are very complex, being in tight relationship with the ECM, tumor, and immune microenvironment, and the accumulation of knowledge on the molecular basis of these interactions may help in diagnosis, prognosis, and overcoming chemoresistance. 

## Figures and Tables

**Figure 1 ijms-23-12415-f001:**
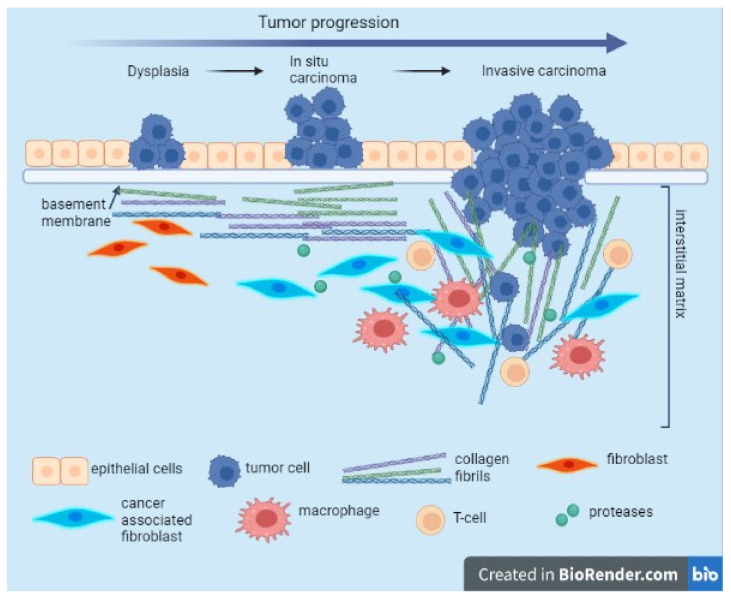
Evolution of collagen fibril organization during tumor progression. During carcinogenesis and cancer development, tumor cells surpass the physical barrier represented by the basement membrane and the interstitial matrix. This event is associated with the enhancement of the activity of the CAFs being able to secrete and reorganize the collagen fibers, increasing the matrix stiffness. Tumor associated macrophages (TAM) and CAFs contribute to collagen remodeling and degradation due to secreted proteases. Modified collagen fibers shift their orientation, allowing migration of invading cancer cells. This image was created with BioRender (https://biorender.com/ (accessed on 30 September 2022)).

**Figure 2 ijms-23-12415-f002:**
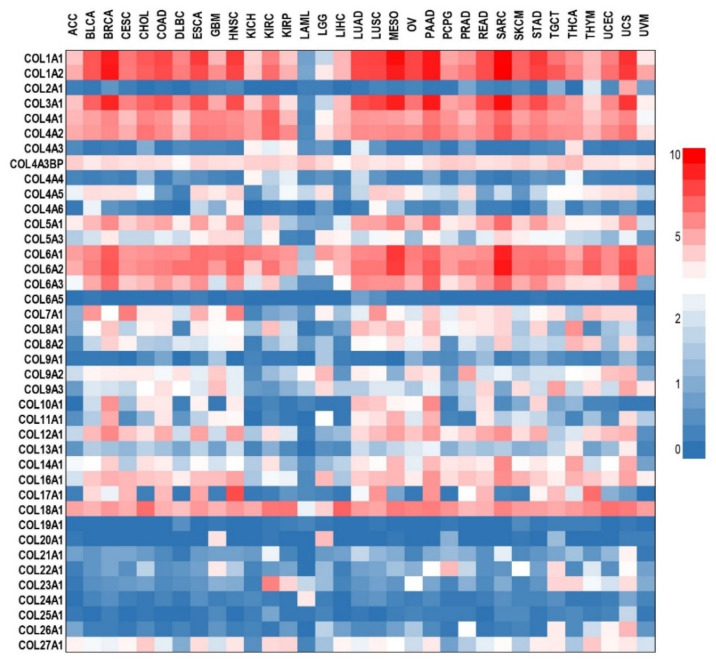
Differential expression analysis of collagens in various cancer types. RNA sequencing expression data from the TCGA and the GTEx projects were analyzed with GEPIA [87] and presented as a heatmap generated in GraphPad Prism 9. Data are expressed as log2(TPM + 1) (TPM- transcripts Per Million). ACC—adrenocortical carcinoma; BLCA—bladder urothelial carcinoma; BRCA—breast invasive carcinoma; CESC—cervical squamous cell carcinoma and endocervical adenocarcinoma; CHOL—cholangiocarcinoma; COAD—colon adenocarcinoma; DLBC—lymphoid neoplasm diffuse large B-cell lymphoma; ESCA—esophageal carcinoma; GBM—glioblastoma multiforme; HNSC—head and neck squamous cell carcinoma; KICH—kidney chromophobe; KIRC—kidney renal clear cell carcinoma; KIRP—kidney renal papillary cell carcinoma; LAML—acute myeloid leukemia; LGG—brain lower grade glioma; LIHC—liver hepatocellular carcinoma; LUAD—lung adenocarcinoma; LUSC—lung squamous cell carcinoma; MESO—mesothelioma; OV—ovarian serous cystadenocarcinoma; PAAD—pancreatic adenocarcinoma; PCPG—pheochromocytoma and paraganglioma; PRAD—prostate adenocarcinoma; READ—rectum adenocarcinoma; SARC—sarcoma; SKCM—skin cutaneous melanoma; STAD—stomach adenocarcinoma; TGCT—testicular germ cell tumors; THCA—thyroid carcinoma; THYM—thymoma; UCEC—uterine corpus endometrial carcinoma; UCS—uterine carcinosarcoma; UVM—uveal melanoma.

**Table 2 ijms-23-12415-t002:** In vivo studies focused on the collagen family.

Collagen Type	Cancer Type	Effect	References
COL1	Pancreatic Cancer	COL1 deletion in myofibroblast pancreatic tumor promotes progression and immunosuppression; CXCR2 and CCR2 inhibition reverses COL1-deletion-induced tumor progression	[113]
	Breast cancer	Promotes mammary tumor initiation and progression; COX2 may be an effective therapeutic target	[100,114,115]
COL7	Lung cancer	Promotes tumor growth and poor prognosis	[113]
COL10A1	Colorectal cancer	Promotes tumor growth and metastasis via epithelial–mesenchymal transition (EMT)	[108]
COL11A1	Breast cancer	Promotes tumor growth and metastasis	[116]
COL12	Breast cancer orthotropic mice model	Promotes metastatic dissemination by decreasing collagen I bundle thickness and tumor stiffness	[44]
COL18A1	Breast cancer	Promotes cancer growth and metastasis by interacting with EGFR/ErbB and activating MAPK/ERK1/2 and PI3K/Akt pathways	[117]

## Data Availability

Not applicable.

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
