# Peer review of "Collagen Family as Promising Biomarkers and Therapeutic Targets in Cancer"

_ijms, 2022, doi:10.3390/ijms232012415_

Round 1
Reviewer 1 Report
The authors introduced the structure and function of collagen. The authors also introduced the role of collagen in cancer. However, there are some concerns of the manuscript that should be addressed before a decision could be made.
Major comments
1. Line 40-47, can these markers be used for early detection?
2. Line 124-128, include some examples supporting pro- and anti-tumor effects of COL4A1.
3. Lines 203-219, no clues for COL4 or COL10 as early markers.
4. Lines 220-226, no clues for COL6 as an early marker.
5. Lines 228-236, no clues for COL1 and COL3 degradation products as early markers.
6. Lines 252-255, examples should be included.
7. Lines 289-294, are there any quantitative data from the literature?
8. Lines 302-305, are these in vitro or in vivo studies? If it’s in vivo studies, what kind of models were used?
Minor comments
1. Line 134, “cancer-associated fibroblasts secreted by COL12”. This sentence is confusing. Revise it.
2. There are lots of grammar errors. For example, Line 31, “Despite the advances in cancer therapy this”, a comma should be included after therapy. Line 46, “some the cancer patients”. Of should be included after some. Line 99. “Pro-a-chain suffers”, could be “Pro-a-chains undergo”. Line 147, “that occurs The” is confusing. Revise it.
Author Response
The authors introduced the structure and function of collagen. The authors also introduced the role of collagen in cancer. However, there are some concerns of the manuscript that should be addressed before a decision could be made.
Major comments
Point 1: Line 40-47, can these markers be used for early detection?
Response 1: Thank you for your question. We introduced a paragraph that summarizes the use of these biomarkers and highlights their utility for the detection of recurrence and postoperative surveillance rather than early detection of new cancers: “For example, even if CA 125 serological test is used in the case of ovarian cancer suspicion, this test has limited utility in early diagnosis of ovarian cancer, due to its low sensitivity and the low specificity in premenopausal women [4]. Moreover, a study on 768 gastric cancer patients showed that 15.4% of patients had increased pre-operative CEA levels, and only 8.7% had increased CA19-9 levels [5], suggesting that these markers are not useful for early detection.
In an attempt to overcome this issue, several studies have shown that an association of CEA, CA19-9, and CA72.4 can be used as triple markers with increased sensitivity, up to 62% in gastric cancer [6].
AFP is the most commonly used biomarker for hepatocellular carcinoma but the level of AFP is also increased in other benign liver diseases. Therefore it is recommended a combination of AFP testing with other factors such as platelets and age, CEA and CA-19-9, and microRNAs for hepatocellular carcinoma screening [7].
PSA is currently used as a biomarker for prostate cancer screening and diagnosis, a low baseline PSA indicating a low risk of developing prostate cancer [8]. German Institute for Quality and Efficiency in Health Care reported this year that PSA testing increases prostate cancer diagnoses and reduces long-term disease-specific mortality, but also results in overdiagnosis and treatment-related harms [9].
Recently, a study on a large cancer patient cohort has highlighted the use of CA19.9 and CEA as prognostic biomarkers, their level being associated with shorter recurrence-free survival (RFS) and overall survival (OS) [10]. In conclusion, the results suggest the utility of these tumor markers for the detection of recurrence and postoperative surveillance rather than early detection of new cancers [11].”
Point 2: Line 124-128, include some examples supporting the pro- and anti-tumor effects of COL4A1.
Response 2: Thank you for your good observations. We introduced a paragraph that contains examples supporting the pro- and anti-tumor effects of COL4A1: “In vitro studies on endothelial cells suggested that tumstatin derived from α3 chain of type IV collagen and canstatin derived from α3 chain of type IV collagen can inhibit angiogenesis by inhibiting human endothelial cell migration and present proapoptotic activity [36-38]. However, in several types of cancer such as breast, pancreas, gastric, and colorectal cancers, fragments of collagen 4 have been identified as possible prognostic markers correlated with increased invasiveness and poor survival most likely due to a higher risk of developing distant metastasis [39-41].”
Point 3: Lines 203-219, no clues for COL4 or COL10 as early markers.
Response 3: Thank you for your observation. We understand the confusion that may arise due to the previous paragraph. In this sense, we introduced the following phrase in the manuscript to clarify the issue: “Plasma levels of different types of collagens were found to be elevated in various malignancies but for the moment only a few molecules that belong to this family can be considered as possible biomarkers for early cancer detection. For example, a recent study suggested that the evaluation of COL3 and MMP-1 levels can be used for the early detection of hepatocellular carcinoma [57]. Similarly, in the case of breast cancer evaluation of the circulating level of COL11A1, COL10A1, and COMP (collagen oligomeric matrix protein) can discriminate between malign and benign disease [58].
However, many studies sustain the association between carcinogenesis and modification in circulating expression of collagen family.”
Point 4: Lines 220-226, no clues for COL6 as an early marker.
Response 4: In accordance with the previous point, we do not sustain the idea that COL6 is an early marker for diagnosis, but rather a potential biomarker for cancer progression and metastasis.
Point 5: Lines 228-236, no clues for COL1 and COL3 degradation products as early markers.
Response 5: In accordance with the above, we do not sustain the idea that COL1 and COL3 are early biomarkers for diagnosis, but rather for cancer prognosis.
Point 6: Lines 252-255, examples should be included.
Response 6: Thank you for your observation. We used that phrase as a general comment to introduce the following subsection “The Role of Collagens in Cancer” where the role of collagen family members in migration, signaling, angiogenesis, invasion, and drug resistance is presented extensively with specific examples, for each type of collagen and its mechanism of action (Table 1).”
Point 7: Lines 289-294, are there any quantitative data from the literature?
Response 7: Thank you for your helpful suggestion. We have introduced quantitative data about COL11 expression in various cancers based on GEPIA data and supported by two other studies on mRNA expression: „However, COL11A1 is well known to be upregulated in several cancers with epithelial cells with a median fold change of 2.54 increasing up to 5.3 in breast cancer, followed by mesothelioma and pancreatic adenocarcinoma [91,92]. As such, COL11A1 was proposed as a tumor marker for prognosis of breast, glioma, head and neck, lung, colorectal, esophagus, gastric, ovarian, pancreatic, salivary gland, and renal cancers [86,93].”
Point 8: Lines 302-305, are these in vitro or in vivo studies? If it’s in vivo studies, what kind of models were used?
Response 8: Thank you for your observation. We specified in the text the type of studies, respectively in vitro, that evaluated the mechanisms beyond COL11 up-regulation in different types of cancer
“According to several in vitro studies Akt, TGF-beta1, B-myb, and Gli1, can up-regulate the transcription of COL11A1 in pancreatic [94], lung [95], and ovarian cancer [96]. The high expression of COL11A1 induces also the expression of molecules such as Twist1 and MMP3, which are related to drug resistance and invasion of cancer [97].”
Minor comments
Point 1: Line 134, “cancer-associated fibroblasts secreted by COL12”. This sentence is confusing. Revise it.
Response 1: Thank you for your observation. We corrected the sentence: “COL12 secreted by cancer-associated fibroblasts (CAFs)”.
Point 2. There are lots of grammar errors. For example, Line 31, “Despite the advances in cancer therapy this”, a comma should be included after therapy. Line 46, “some the cancer patients”. Of should be included after some. Line 99. “Pro-a-chain suffers”, could be “Pro-a-chains undergo”. Line 147, “that occurs The” is confusing. Revise it.
Response 2: Thank you for your observations. We verified and made all necessary corrections.
Reviewer 2 Report
The review is very good, it provides a lot of information on how collagen can be used as a cancer biomarker or in future cancer therapies. However, the two figures presented are confusing and uninformative. In Figure 1, the mechanism should be indicated by arrows or other visual aids. Figure 2 is not very informative, there is no mention of how was done the heatmap, was a bioinformatics program used?
3.2. Plasma Levels of collagen in Cancer Patients. (This form is correct)
I consider that the work should be accepted with these minor corrections.
Author Response
The review is very good, it provides a lot of information on how collagen can be used as a cancer biomarker or in future cancer therapies.
Point 1. However, the two figures presented are confusing and uninformative. In Figure 1, the mechanism should be indicated by arrows or other visual aids. Figure 2 is not very informative, there is no mention of how was done the heatmap, was a bioinformatics program used?
Response 1: Thank you for your kind words and appreciation.
Regarding Figure 1, we added some arrows and information indicating the tumor progression. Also, we completed the legend adding more information regarding collagen organization during cancer development
„Figure 1. Evolution of collagen fibrils organization during tumor progression. During carcinogenesis and cancer development tumor cells surpass the physical barrier represented by the basement membrane and the interstitial matrix. This event is associated with the enhancement of the activity of the CAFs able to secrete and reorganize the collagen fibers increasing the matrix stiffness. Tumor-associated macrophages (TAM) and CAFs contribute to collagen remodeling and degradation due to secreted proteases. Modified collagen fibers shift their orientation, allowing migration of invading cancer cells. This image was created with BioRender (https://biorender.com/).”
Regarding Figure 2, we completed the legend specifying the data source and the software used to generate the heatmap „RNA sequencing expression data from the TCGA and the GTEx projects were analyzed using GEPIA [87] and presented as a heatmap generated in GraphPad Prism 9”.
Point 2. 3.2. Plasma Levels of collagen in Cancer Patients. (This form is correct)
Response 2: Thank you for your observation. We corrected the title.
I consider that the work should be accepted with these minor corrections.
Round 2
Reviewer 1 Report
The authors addressed most of the comments. However, comments #6 has not been addressed yet. And there are still some grammar errors. See below.
Line 122, “undergo” should be “undergoes”.
Line 127, “undergo” should be “undergoes”.